# Impact of Pruritus on Sleep Quality of Hemodialysis Patients: A Systematic Review and Meta-Analysis

**DOI:** 10.3390/medicina55100699

**Published:** 2019-10-17

**Authors:** Inayat Ur Rehman, Tahir Ali Chohan, Allah Bukhsh, Tahir Mehmood Khan

**Affiliations:** 1Department of Pharmacy, Abdul Wali Khan University Mardan, Mardan 23200, Pakistan; inayat.rehman@awkum.edu.pk; 2Institute of Pharmaceutical Sciences, University of Veterinary and Animal Sciences Lahore, Lahore 5400, Pakistan; 3School of Pharmacy, Monash University Malaysia, Bandar Sunway 47500, Selangor Darul Ehsan, Malaysia; allah.bukhsh@monash.edu

**Keywords:** end stage renal disease, hemodialysis, CKD-associated pruritus, sleep, chronic kidney disease, systematic review, meta-analysis

## Abstract

*Background and objectives:* Chronic kidney disease (CKD)-associated pruritus is a common and disturbing condition which has a negative impact on sleep quality, as well as overall health-related quality of life of patients receiving hemodialysis. To date, no systematic review has been undertaken, and there is a lack of concise evidence that statistically quantifies the impact of pruritus based on published data. *Materials and Methods:* A systematic search was done for original articles published in peer-reviewed English journals from database inception on 20 December, 2018, in the following databases: PubMed, MEDLINE, EMBASE, Ovid, CINHAL, ProQuest, and Scopus. *Results:* A total of 9217 research articles were identified. After removal of duplicates and screening for titles and abstracts, 28 articles were selected. The prevalence of disturbed sleep was 4–94%, while the pooled proportion on random effect in the study was 40% (95% CI = 0.30 to 0.49); I^2^ = 99.8%. However, the prevalence of disturbed sleep quality and quantity due to pruritus was 9–76%, and the pooled proportion on random effect in the study was 50% (95% CI = 0.37 to 0.64); I^2^ = 99.8%. *Conclusions:* Patients undergoing hemodialysis who are affected by CKD-associated pruritus have a higher chance of experiencing sleep disturbances. The prevalence of disturbed sleep due to CKD-associated pruritus was found to be 9–76% in the included studies for patients receiving hemodialysis therapy.

## 1. Introduction

Chronic kidney disease (CKD) is a major public health dilemma worldwide [1]. CKD-associated pruritus (CKD-aP) contributes to patient complications [2,3], and is a common and troublesome complication experienced by CKD patients undergoing hemodialysis which has a major and distressing effect on their health-related quality of life (HRQOL) [4,5]. The prevalence of CKD-aP ranges from 22–84% among patients undergoing hemodialysis [5,6,7], and is more frequent at night and affects patient’s sleep and moods [8]. CKD-aP may be episodic or constant, generalized or localized, and its intensity may vary from mild to severe [9]. It is quite often overlooked in hemodialysis patients, leading to worsening itchiness, which then results in poor HRQOL, and sleep disturbances in patients with chronic renal failure [10,11,12]. Sleep quality affected by CKD-aP varies in different studies, ranging from 11–26% [13,14,15,16]. Previous literature established that CKD-aP leads to compromised physical and mental wellbeing, and poor HRQOL in CKD patients [14,17]. Approximately 20–50% of patients report that CKD-aP has a negative effect on their lives [18,19]. Although a lot of treatment options are available for the management of CKD-aP, treatment is difficult due to the refractory nature of the disease, resulting in poor and compromised HRQOL, including sleep [14].

Due to the impact of CKD-aP on patients, and the challenges associated with its treatment, it would seem imperative to quantify the likelihood of CKD-aP among patients undergoing hemodialysis. Moreover, it will be of great worth to explore the impact of CKD-aP on the sleep quality of patients undergoing hemodialysis. To date there is no systematic review and a lack of concise evidence that statistically quantifies the impact of CKD-aP based on published data. Our objective is to therefore evaluate the prevalence of sleep disturbance and CKD-aP among hemodialysis patients. Findings can be used to ensure that appropriate and timely measures can be planned to improve the HRQOL of these patients, a factor which is often overlooked in clinical settings [20]. Therefore, our objective in this systematic review was to evaluate the prevalence of sleep disturbance in CKD-aP patients on hemodialysis.

## 2. Methods

A systematic review was conducted to explore the impact of CKD-aP on the sleep quality of patients undergoing hemodialysis. The Preferred Reporting Items for Systematic Reviews and Meta-Analyses guidelines were followed to identify potential research articles from evidence-based scientific literature (PRISMA Statement) [21]. The protocol for this systematic review has been registered on Prospero, and can be accessed online at PROSPERO 2016: CRD42016050080.

### 2.1. Study Identification

PubMed, MEDLINE, EMBASE, Ovid, CINHAL, ProQuest, and Scopus were searched for potential papers from database inception to 20 December, 2018.

### 2.2. Search Strategies

The strategic search terms used medical subject headings (MeSH) and keywords, and the following text terms were combined with Boolean operators: (uremic* OR Skin Disorder OR Rash* OR Itch* OR Pruritus OR Uremia*) AND (Sleep* OR Disturb* sleep OR insomnia OR sleep quality OR lack of sleep OR no sleep) AND (Kidney OR Renal OR CKD OR ESRD OR end stage renal disease OR chronic kidney failure OR chronic kidney disease OR renal impair* OR Renal insuff*) AND (dialysis OR hemodialysis).

### 2.3. Study Selection

#### Inclusion Criteria

Original research articles using observational or experimental designs, published in peer-reviewed English journals and exploring the impact of CKD-aP on sleep quality among CKD patients undergoing hemodialysis.

#### Exclusion Criteria

All systematic reviews, case reports, advertisements, thesis, opinions, letters to the editor, and qualitative studies, were excluded. In addition, studies which reported on, or data on any other form of pruritus due to factors other than CKD, were excluded.

### 2.4. Data Extraction and Quality Assessment

Data were extracted using a standardized extraction form designed for this systematic review. The following information was extracted: author’s name, publication year, study design, country, respondent’s information, inclusion/exclusion criteria for the selection of respondents, sample size, statistical analysis used, and outcome measurements of the impact of CKD-aP on sleep quality and quantity. The Cochrane risk of bias [22] criteria was used to assess the risk of bias for randomized clinical trials (RCTs), while the Newcastle Ottawa Scale (NOS) [23] was selected for quality appraisal of observational studies. Any disagreement was resolved through discussion.

### 2.5. Data analysis

For the risk of bias, RevManger 5.3® was used to generate a graphical presentation of the included trials. A meta-analysis was performed using STATA version 14 using the random effects model. All *p*-values were set at <0.05 with 95% confidence intervals. Subgroup analysis was performed to address heterogeneity. I^2^ statistic was used to interpret the heterogeneity at a confidence interval of 95% among the included studies.

## 3. Results

### 3.1. Study Selection

A total of 9217 related research articles were identified. After removal of duplicates, the final count reduced to 8054. After reviewing titles and abstracts, 777 were selected, and from these, 28 articles were included in this systematic review after the application of the inclusion and exclusion criteria [13,14,15,16,18,24,25,26,27,28,29,30,31,32,33,34,35,36,37,38,39,40,41,42,43,44,45] (Figure 1).

### 3.2. Study Characteristics

The study characteristics are shown in Table 1. Patients undergoing hemodialysis who developed CKD-aP experienced delays in sleep, awakenings due to CKD-aP, and nocturnal awakenings (Table 1).

### 3.3. Disturbed Sleep Quality in Patients Undergoing Hemodialysis

Sleep disturbance among hemodialysis patients in the selected studies [13,14,15,16,18,24,25,26,27,28,29,31,32,33,34,35,36,44,45,46] the minimum prevalence of sleep disturbance was 4% [27], while the study with the highest prevalence of sleep disturbance had a maximum prevalence of up to 94% [31]. Quantitative analysis using the pooled proportion revealed that the likelihood of sleep disturbance among the selected studies was 40% (95% CI = 0.30 to 0.49); I^2^ = 99.8% (shown in Figure 2). 

#### Sub Group Analysis

The pooled proportion on random effects revealed the prevalence of sleep disturbance in cross sectional studies was 32% (95% CI = 0.19 to 0.45); I^2^ = 99.8%; while in prospective observational studies it was 49% (95% CI = 0.33 to 0.65), and in clinical trials it was 74% (95% CI = 0.52 to 0.96). The pooled proportion on random effect found prevalence of sleep disturbance for studies conducted in Asia was 50% (95% CI = 0.38 to 0.62) and the heterogeneity was 99.0%; whereas the pooled proportion on random effect for studies conducted in Europe was 25% (95% CI = 0.19 to 0.32) and the heterogeneity was 80.1%. 

### 3.4. Impact of CKD-aP on the Sleep Quality of Patients on Hemodialysis

For estimation on the impact of CKD-aP on sleep quality, selected studies [14,25,26,27,28,35,38,39,41,42,43,44,45], showed the overall, the pooled proportion on random effect in studies was 50% (95% CI = 0.37 to 0.60); I^2^ = 99.3% (shown in Figure 3).

#### Sub Group Analysis

The results of the subgroup analysis on the basis of geographical regions among selected studies [14,25,26,27,28,35,42,43,44] revealed that the overall pooled proportion of sleep disturbance due to CKD-aP was 50% (95% CI = 0.32 to 0.69); I^2^ = 98.2% in Asia and 54% (95% CI = 0.465 to 0.62) in Europe. Subgroup analysis based on sample size showed a pooled proportion of 42% (95% CI = 0.15 to 0.70); I^2^ = 97.3% for sample size less than 200, while for sample size more than 200 showed a pooled proportion of 55% (95% CI = 0.38 to 0.71); I^2^ = 99.0%.

### 3.5. Quality Assessment and Risk of Bias of Included Studies

The quality assessment of observational studies was assessed using NOS [23]. The mean NOS score for the included studies was 6.52 ± 0.511, with a range of 6–7. Ten studies [15,16,18,24,25,26,27,28,30,32] have a score of 6, which grades as fair, while for 14 studies [13,14,33,36,38,39,41,42,43,44,45,46,47,48] achieved a score of 7, which is graded as good. For RCTs, Makhlough (2010) [29] was comparatively better in quality than the other three trials [31,37,40] which have one or more chances of a high risk of bias (Figure 4 and Figure 5).

## 4. Discussion

This systematic review and meta-analysis is perhaps the first systematic and quantitative assessment to assess the impact of CKD-aP on sleep quality among CKD patients undergoing hemodialysis. Results from the meta-analysis revealed the proportion of hemodialysis patients suffering from sleep disturbance, regardless of whether they have CKD-aP or not, was 40% (95% CI = 0.30 to 0.49); I^2^ = 99.8%, while in patients suffering from CKD-aP, 50% of them were observed to have sleep disturbances with a pooled proportion of 50% (95% CI = 0.37 to 0.64); I^2^ = 99.8%. This emphasizes the need to routinely assess CKD-aP and sleep quality. This is to allow suitable and timely interventions to be carried out to address these problems, which are very frequent among CKD patients on hemodialysis.

The heterogeneity in the prevalence data may be due to the different study design, selection of participants (e.g., study populations, races, and sample sizes), and the definition of CKD-aP in the included studies. Another meta-analysis on the prevalence of CKD-aP among adult hemodialysis patients also previously reported high heterogeneity in the subgroup analysis [49].

Studies showed a high prevalence of CKD-aP with 85.4% [50] and 53.4% [51], and CKD-aP led to sleep disturbances in 33.8% of patients [51]. Furthermore, patients with CKD-aP have a poorer quality of sleep or more serious depression as compared to patients without CKD-aP [52,53,54].

Therefore, from a clinical practice point of view it is important to assess patients for the severity of CKD-aP and intervene therapeutically to improve the HRQOL and overall wellbeing of the patients. CKD-aP is often overlooked by nephrologists, primary care physicians, and other health care professionals [20] and this negligence can be curtailed by training the nephrologist on how to interpret tools to quantify the severity of CKD-aP and sleep. Moreover, it is essential to provoke awareness among physicians, nephrologists, and researchers for formulations of coherent and suitable therapeutic strategies and guidelines to manage CKD-aP. Indeed, CKD-aP should be treated with equal importance as other symptoms because scratching to relieve CKD-aP may impair sleep quality as well as other QOL aspects of the patients, and this can cause other medical related complications like secondary infections.

## 5. Strength and Limitation

There was high heterogeneity in the pooled analysis; this could be due to varied sample size, population characteristics, race, difference in study designs, diverse tools/questionnaire (validated and self-designed) used for assessment of CKD-aP as well as sleep assessment and the definition of CKD-aP in the included studies. Despite performing sub-group analysis, the heterogeneity remained high and that is the major limitation of this study.

## 6. Conclusions

In conclusion, sleep quality is disturbed to a great extent by CKD-aP in patients receiving hemodialysis. The prevalence of disturbed sleep due to CKD-aP was found to be 9–76% in the included studies; establishing evidence of impaired sleep quality. More research is needed to establish suitable data using specific tools for determining CKD-aP and sleep quality for quantitative analysis. Among health care providers, CKD-aP should be treated with equal importance as other complications to improve patient’s HRQOL, and to relieve itching to avoid secondary infections due to persistent scratching.

## Figures and Tables

**Figure 1 medicina-55-00699-f001:**
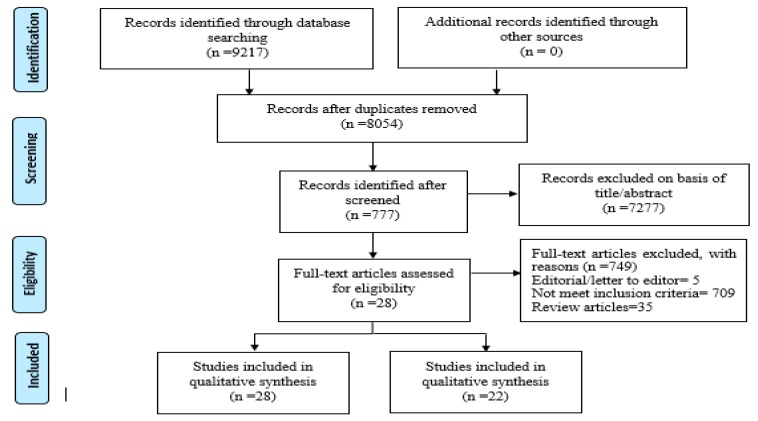
Preferred Reporting Items for Systematic Reviews and Meta-Analyses (PRISMA) flow diagram of study selection.

**Figure 2 medicina-55-00699-f002:**
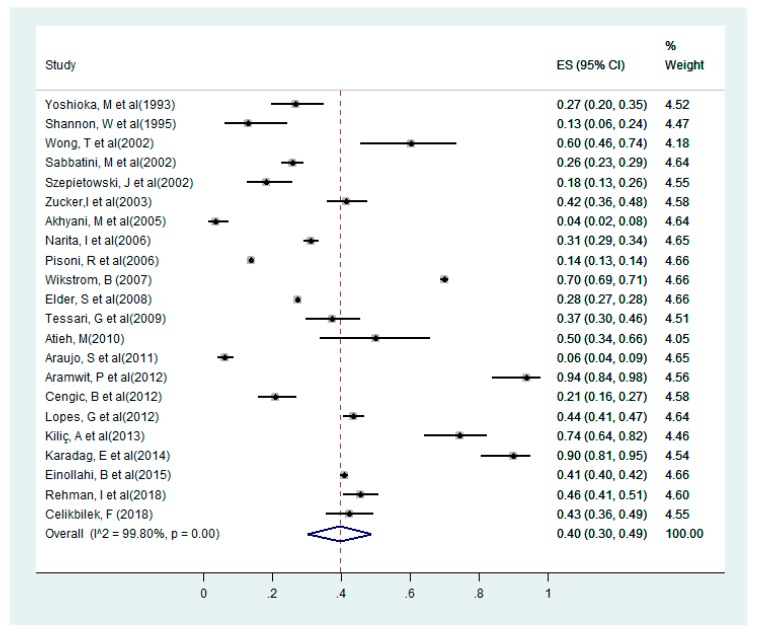
Forest plot for sleep disturbance among dialysis patients in the selected studies.

**Figure 3 medicina-55-00699-f003:**
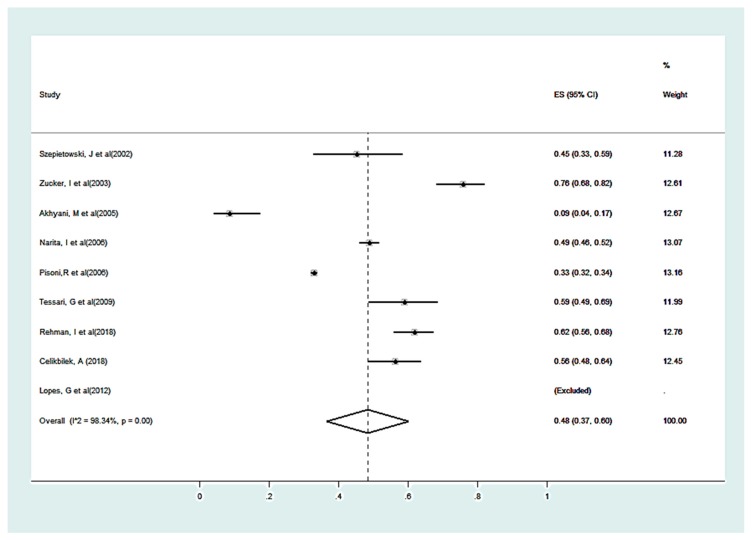
Forest plot for patients with sleep disturbances due to pruritus among selected patients.

**Figure 4 medicina-55-00699-f004:**
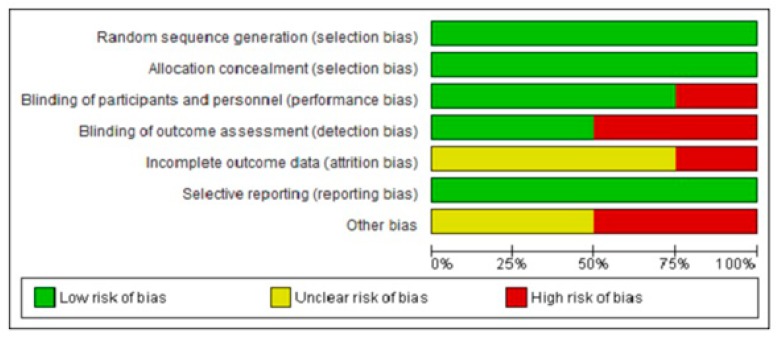
Risk of bias graph: Review authors’ judgments about each risk of bias item presented as percentages across all included.

**Figure 5 medicina-55-00699-f005:**
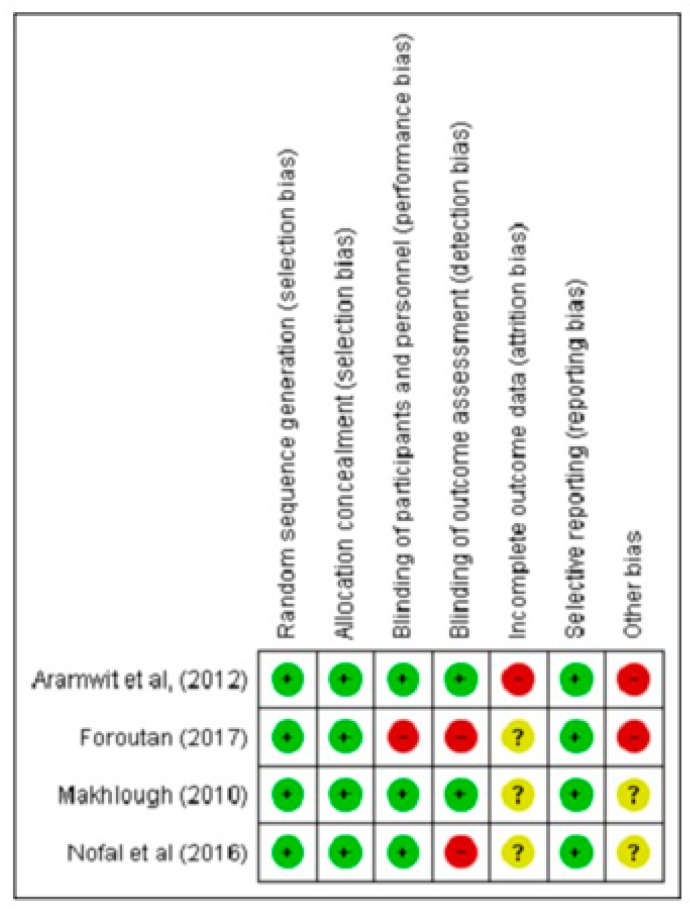
Risk of bias summary: Review authors’ judgments about each risk of bias item for each included Randomized Clinical Trials.

**Table 1 medicina-55-00699-t001:** Recent studies examining the association of CKD-aP with sleep quality of dialysis patients. Cs: cross sectional study, OS: observational study, RCT: randomized controlled trial, HD: hemodialysis, PD: peritoneal dialysis.

Authors	Year	Study Design	Sample	Drop Out	Result (Sleep Disturbed Due to Pruritus)
Yoshioka and Ishii et al. [16]	1993	CS	127	0	26.8% of patients
Walker et al. [15]	1995	CS	64	10	Delayed sleep onset in 8% of patients, nocturnal awakenings due to pruritus in 25%
Wong et al. [24]	2002	CS	43	0	60% of patients
Sabbatini et al. [18]	2002	CS	737	43	19.9% of patients
Szepietowski et al. [25]	2002	CS	130	0	45.3% of patients
Zucker et al. [26]	2003	CS	264	45	61% of patients awakened frequently and 44% occasionally
Akhyani et al. [27]	2005	CS	167	0	6 patients
Pisoni et al. [44]	2006	CS	18,801	0	45% of patients
Wikstrom [46]	2007	CS	6137	0	72% of patients
Elder et al [45]	2008	CS	17,034	5683	52.2% of patients
Tessari et al. [28]	2009	CS	169 (139 HD; 30 PD)	0	59.1% of patients
Mathur et al. [30]	2010	OS	103	0	Type B (2.9% of patients have a problem falling sleep because of pruritus); while in Type C (3.0% often have, problems falling sleep because of pruritus)
Araujo et al. [13]	2011	CS	400	0	11.0% of patients
Aramwit et al. [31]	2012	RCT	50	3	At enrollment day: Sleep score = 43.41 ± 22.08; after 6 weeks with using sericin cream sleep score = 52.74 ± 18.22
Cengic et al. [32]	2012	CS	200	0	28% of patients
Lopes et al. [14]	2012	CS	980	0	Mildly bothered by pruritus: 24.38%; while Severely: 19.38%
Kiliç Akça [33]	2013	Non-RCT	86	8	32/38 (intervention group) and 32/40 (control group)
Karadag et al. [34]	2013	OS	70	0	90% of patients
Narita et al. [35]	2006	OS	1773	0	70% of patients
Einollahi et al. [36]	2015	CS	6979	0	60.6% of patients
Rehman et al. [42]	2018	CS	354	0	162 of patients
Celikbilek [43]	2018	CS	200	0	10.6% of the patients almost always, while 56.3% of patients occasionally affects sleep

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
