# Peer review of "Impact of Pruritus on Sleep Quality of Hemodialysis Patients: A Systematic Review and Meta-Analysis"

_medicina, 2019, doi:10.3390/medicina55100699_

Round 1
Reviewer 1 Report
The authors performed a metaanalysis on the role of pruritus in sleep deprivation in dialysis patients. I found the topic of great relevance. The methodology, that was used by the authors is sound. However, I have some concerns regarding the entire manuscript.
Based on the title, the authors wanted to find an answer to the question, whether pruritus causes (and to what extend) sleep problems in dialysis patients? However, based on this data presentation, the question remains unanswered as it is clear overlap between the prevalence of sleep problems in the entire dialysis population and those patients 40% (95% CI = 0.30 to 0.49), who suffered from pruritus(50% (95% CI = 0.37 to 0.64)). This is in part due to huge heterogeneity of included studies, but in my opinion, due to lack of some additional subanalyses. I have following questins/suggestions to provide further data on the major aim of this metaanalysis:
Did the authors look for the differences in sleep problems between studies, in which sleep problems was a primary endpoint versus those studies where it was one of many secondary endpoints? Did the authors look for the prevalence of sleeping problems in those patients who do not suffer from pruritus? What is the relative risk of having sleep problems when you suffer from pruritus versus those dialysis patients who are free from this symptom? A little bit more attention should be put on the definition of the phrase "sleep disturbance". How did you define "sleep quality" and "sleep quantity"? In my opinion, you have only looked for prevalence of sleep disrtubances. Did your studies used REM/nonREM sleep assessemnts? If yes, please provide analysis, if not, please change theFigure 1 is misleading. Based on this figure, no study was included in the metaanalysis. In addition - Additional records indetified through other sources is also 0 - it is fine, but nothing is written in the methodology, how did you look for it? No info is aso provided on screening procedures - how did you exclude 7277 records - by title?
Figure 2 - please change "disturance" to "disturbance". The same regarding figure 3 : change "disturbane" to "disturbance"
Reviewer 2 Report
It is a good undertaking to bring data on this topic.
I see very minor spelling changes that need address ing. Otherwise a good review.
- The heading on figure 2 & 3 need to be corrected for spelling
Round 2
Reviewer 1 Report
The authors analysed sleep problems in patients on dialysis. The topic is relevant and interesting. Unfortunately, the authors did not respond satisfactorily to any major query raised during the previous review round. The metaanalysis is rather superficial, without any critical analysis of published data describing limitation of analysed studies. It is even not defined, what "sleep disturbances" mean.
figure 1 has been modified, but it is still incorrect - please look at the lowest row.
Author Response
24th September 2019
Editor in chief
MDPI Medicina
Dear Sir,
Response to reviewers’ comments for manuscript ID: medicina-566515; entitled: “Impact of Pruritus on Sleep Quality of Hemodialysis Patients: A Systematic Review and Meta-Analysis”.
Reviewer 1
Comment: The authors analyzed sleep problems in patients on dialysis. The topic is relevant and interesting. Unfortunately, the authors did not respond satisfactorily to any major query raised during the previous review round. The meta-analysis is rather superficial, without any critical analysis of published data describing limitation of analyzed studies. It is even not defined, what "sleep disturbances" mean.
Response: Thank you for your comment, I tried to answer the comments raised in revision 1. As the data is not primary data and is collected from published papers, that is why superficial analysis is performed which is considered as a limitation. While for the sleep disturbance, the papers reported disturbed sleep as sleep disturbances.
Comment: Figure 1 has been modified, but it is still incorrect - please look at the lowest row.
Response: Thank you for your comment, the figure 1 has been modified and incorporated in the manuscript.
Regards
Dr. Tahir Mehmood Khan
Institute of Pharmaceutical sciences
UVAS, Lahore
tahir.khan@uvas.edu.pk

Round 3
Reviewer 1 Report
None